**communications** engineering

# The effect of device geometry on the performance of a wave energy converter
Emma C. Edwards [1,2] ✉, Craig Whitlam[3], John Chapman[3], Jack Hughes [3], Bryony Redfearn[3],
Scott Brown [2], Scott Draper [4,5], Alistair G. L. Borthwick [2,6], Graham Foster [3], Dick K.-P. Yue [7],
Martyn Hann[2] & Deborah Greaves [2]

Wave energy presents an excellent opportunity to add much-needed diversification to the global
renewable energy portfolio. However, a competitive levelised cost of electricity for wave energy
conversion devices is yet to be proven. Here, we optimise the geometry of a wave energy device to
maximise power while also minimising the power take-off reaction moments. Using theory, numerical
modelling and optimisation techniques, we show that by including minimisation of reaction moments
in the optimisation, instead of only maximisation of power, it is possible to substantially lower the
design loads while maintaining high efficiency. Using the underlying physics of how geometry affects
the wave-structure interaction, we explain the resulting performance of these new designs for wave
energy converters. We examine the resulting geometries for practicality, including performance over a
wide range of sea states, motion requirements, and performance in a real sea-state off the coast of
Scotland, United Kingdom. Comparing against the single shape which extracts the theoretical
maximum power, the optimal shapes found in our study extract almost as much power (12% less) with
substantially less moment (reduced by up to 35%), revealing a promising direction for wave energy
development.

As the world transitions to Net-Zero, it will be necessary to rely on energy systems predominantly powered by renewable energy sources. A diverse set of energy generation technologies provides resilience in an energy system[1], so while there is certainly enough solar and wind resource to provide for all our global energy needs, over-reliance on a small set of technologies and generation sources leads to fragility. There is a vast amount of clean energy available in ocean waves—in fact, there is theoretically sufficient power in ocean waves to satisfy the entire global energy requirement[2]. Wave energy could therefore play an important role in diversifying future energy systems because, although correlated, wave energy is not concurrent with wind and thus expands the time window when intermittent renewable generation is available[1].

Wave energy converter (WEC) developers use multiple strategies to extract energy from ocean waves, including the use of point absorbers, overtopping devices, oscillating water columns and attenuators. In the present study, we consider a top-hinged WEC, shown in Fig. 1[3], which consists of an absorber attached to a fixed reference point above the water surface via a hinged rigid arm. In response to waves, the WEC rotates, or

pitches, about the hinge, where the power conversion equipment, called the power take-off (PTO), is located. There are several key advantages of this type of WEC. Firstly, the primary absorber of a top-hinged WEC can be lifted during storms, enabling it to avoid immersion in extreme sea states, unlike most other WEC devices (see e.g. ref. 4). Secondly, the location of PTO equipment above the water surface eliminates the need for an effective submerged seal on the moving mechanism. Additionally, this type of WEC can passively adjust to tidal variation and yaw to face the direction of the dominant incident waves, increasing overall energy yield. Moreover, this type of WEC can be attached to other offshore infrastructure, such as a fixed or floating offshore wind platform, facilitating cost reduction through shared infrastructure and maintenance.

While the potential of wave energy has been realised since at least the 1970s when Salter proposed a concept for a WEC[5], wave energy does not yet have a levelised cost of energy (LCOE) that is competitive with other renewable energy resources such as solar and wind energy. Critical to making wave energy a competitive technology is engineering consensus or convergence on an optimal design. There have been hundreds of ideas for

[1]Department of Engineering Science, University of Oxford, Oxford, UK. [2]School of Engineering, Computing and Mathematics, University of Plymouth, Plymouth, UK. [3]Marine Power Systems, Swansea, Wales, UK. [4]Oceans Graduate School, The University of Western Australia, Perth, WA, Australia. [5]Department of Civil, Environmental and Mining Engineering, The University of Western Australia, Perth, WA, Australia. [6]School of Engineering, University of Edinburgh, Edinburgh, Scotland, UK. [7]Department of Mechanical Engineering, Massachusetts Institute of Technology, Cambridge, MA, USA. ✉e-mail: emma.edwards@eng.ox.ac.uk

**Fig. 1 | A top-hinged wave energy converter (WEC), courtesy of Marine Power Systems.** Note that the width of the absorber is of order 10–20 m.

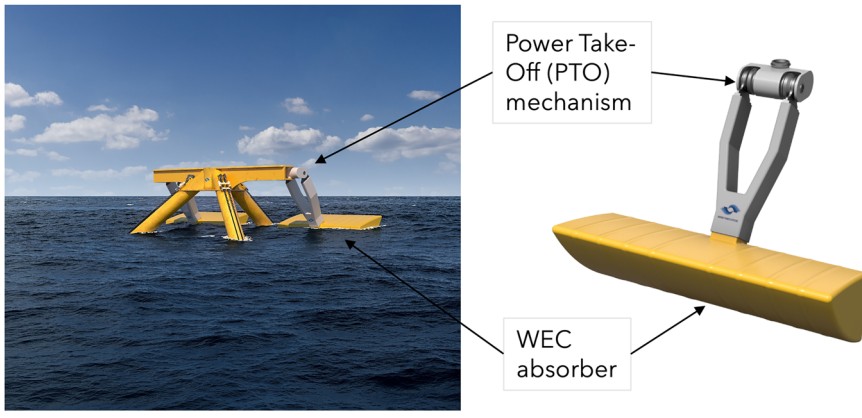

WECs, indicating a lack of such convergence[6,7]. Geometry of a floating body can substantially affect the wave-structure interaction and resulting body response and loads, and so geometry optimisation provides an excellent opportunity to improve the performance of a WEC[8,9]. Many WEC optimisation studies have focused predominantly on maximising power[10,11], and if cost is considered, material mass is typically used as a proxy (i.e.[12–14]). However, for all WEC devices, the PTO, which converts the relative motion between the WEC and the reference platform to electricity, can incur up to 50% of the total capital expenditure[15]. A large mean, peak, or mean-to-peak ratio of reaction force leads to high duty and structural fatigue of the system, so lowering the force/moment will ultimately reduce the cost[16,17]. While some optimisation studies have considered PTO force by maximising the ratio of power to PTO force[7,17], or by minimising fatigue[18], no study has looked at how WEC geometry affects PTO force/moment or how geometry can be exploited to minimise PTO moment. Given the fundamental role of geometry in wave-structure interaction, this represents a substantial gap in current knowledge.

To address this gap, we develop a multi-objective optimisation, where the objective functions are: (i) to maximise extractable power, and (ii) to minimise moment on the PTO mechanism. However, it is not feasible to perform a geometry optimisation over all types of WECs, due to their different working principles, and so we focus on a single WEC to demonstrate the methodology and gain insight into the effect of geometry on the hydrodynamic performance of the device. Our study focuses on a top-hinged WEC due to its advantages (listed above). Our framework yields sets of optimal shapes, and we explain the shape features using the underlying physics of wave-body interactions. We are then able to gain general insight into why the optimal shapes look the way they do. By comparing the WEC shapes against the single shape which extracts the theoretical maximum power (i.e. the shape which would result if maximising power was the only defined objective), our study indicates that the optimal shapes extract almost as much power (~12% less) with substantially less moment (reduced by up to ~35%), revealing a promising direction for WEC development.

## Results
### The discovery of optimal shapes
The problem setup and associated parameters are shown in Fig. 2a. The WEC consists of a floating body on the water surface (the WEC absorber) and a rigid arm, hinged at a fixed point $O$, restricting its motion to pitch (rotation about point $O$) only. The WEC is uniform in $y$, with width $l$. The front and rear faces are defined by curves $c_1$ and $c_2$, respectively, which, along with the length components shown in Fig. 2a, are the parameters to be optimised. This method to define geometry enables many, very general geometries to be described by relatively few parameters.

Sensitivity studies were performed to ascertain how different geometric parameters affect power and moment, from which it was determined to set the lengths of the rigid arm, $s_1$ and $s_2$, and the draft, $h$, to be constant. The rigid arm length parameters were set constant for practical constraints,

usually determined by the support structure. Draft was set to a constant value because both the power and moment were found to increase monotonically as depth increased. Having established this relation, the geometric parameters considered in the optimisation are $r_1$, $r_2$ and curves $c_1$ and $c_2$. These parameters were found to influence power and moment in a meaningful (non-monotonic) way, and most clearly demonstrated the effect of body geometry on performance. The PTO mechanism is assumed to be a simple linear damper in pitch located at point $O$.

Defining a rigorous framework for the optimisation is a key contribution of this study. In this study, we assume the incident wave amplitude to be small and the fluid to be ideal, allowing linear potential flow theory to be used, which is a good approximation for most operating conditions. This assumption is discussed further in the following section. Initially, we assume a single monochromatic unidirectional wave incident from the left, with frequency $\omega = \omega_r$ and wavenumber $k = k_r$. It has been shown (see 'Methods') that to maximise the extractable power for a WEC moving in a single degree of freedom, the device should be in resonance and the PTO damping coefficient should be equal to the radiation damping coefficient at resonance. Therefore, these two criteria are enforced in our study, ensuring the device extracts the maximum possible power at a given frequency. As motivated in the introduction, we have defined a multi-objective optimisation, summarised in Fig. 2b, c, whereby the objective functions are to (i) maximise extractable power, nondimensionalised as $k_r W$, where $W$ is extractable power over incident power per unit crest length, and (ii) minimise PTO moment, nondimensionalised as $|\widetilde{F_5}| = F_5/(\rho g s_1 l A^2)$, where $\rho$ is density of water. To perform this optimisation, we use a Multi-Objective Evolutionary Algorithm[19], and the result is a set of solutions. Subsequently, we examine motion amplitudes for the optimal shapes and the sensitivity of the performance of the shapes to wave frequency and incident wave angle. We perform three optimisations, corresponding to three widths of the absorber, to determine how the optimal geometries depend on changing width, which is of particular interest since it has been observed in the offshore wind sector that larger rotors reduce LCOE. This methodology shares similarities to that of ref. 8, which optimises the geometry of an axisymmetric point-absorber WEC. In particular, a similar geometry parameterisation is used to define the geometries of the WECs in both studies, and a similar optimisation procedure is used. The different underlying physics between the two problems necessitates different frameworks and, as a consequence, different resulting shapes and conclusions.

The resulting Pareto Fronts are shown in Fig. 3. As depicted in inset 1 of Fig. 3, it can be shown (see 'Methods') that for a device uniform in $y$ and restricted to motion in one degree of freedom, the extractable power and PTO moment can be expressed in terms of $A^+$ and $A^-$, the far-field amplitudes of waves generated by forcing the WEC to move with unit amplitude in otherwise calm water. $A^+$ is the wave in the direction of the incident wave ($x = \infty$, yellow) and $A^-$ is the wave in the direction opposite to the incident wave (at $x = -\infty$, pink). It can be shown (see 'Methods') that to minimise PTO moment, $A^-$ (pink) should be minimised. To maximise

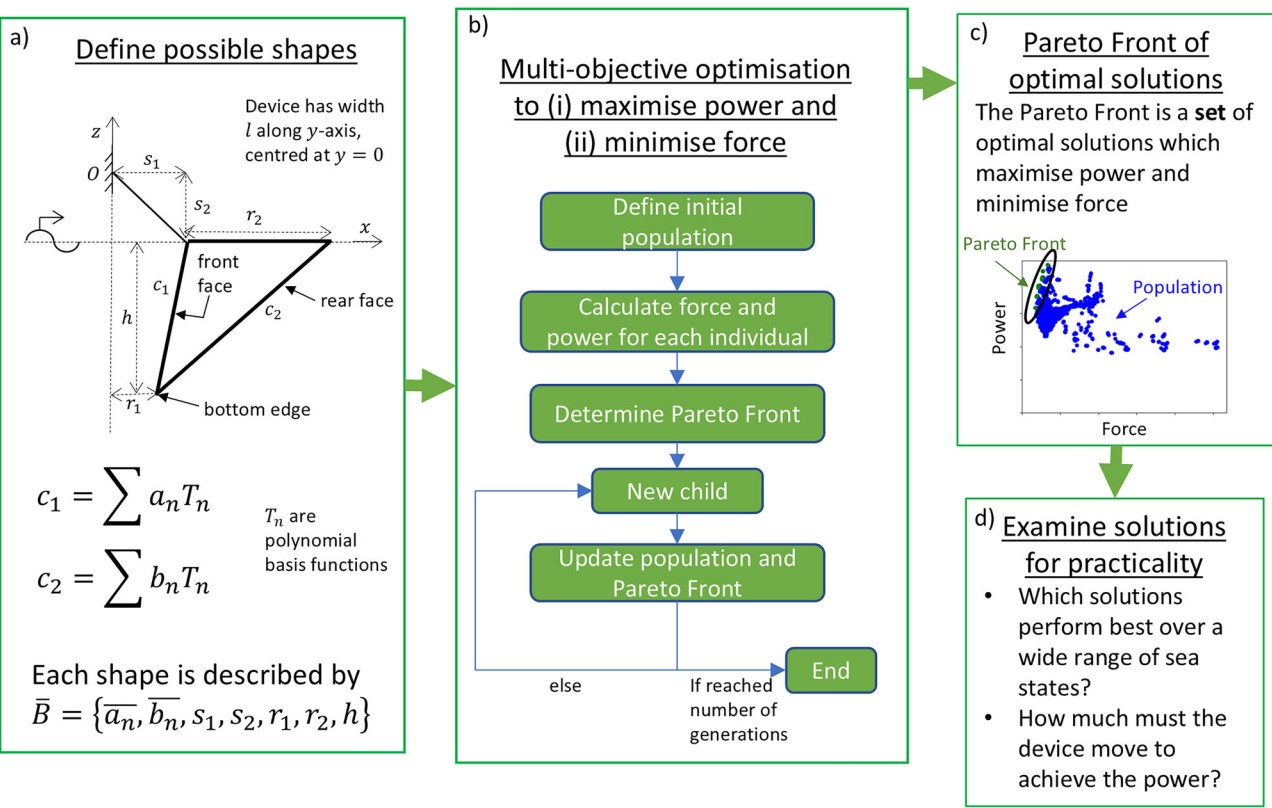

**Fig. 2 | Flow chart showing the problem setup and methodology to find optimal geometries of a wave energy converter. a** Problem setup, with defined geometric shape parameters; **b** Multi-objective optimisation flow chart; **c** Example of the resulting population (blue) and Pareto Front (green); **d** Explanation of steps taken to examine the practicality of the resulting set of optimal shapes from the optimisation.

power, $A^+$ (yellow) should be minimised and $A^-$ (pink) should be maximised. Therefore, there is an unchallenged goal of minimising $A^+$ (yellow) but competing goals of minimising vs. maximising $A^-$ (pink) between the two objective functions.

Looking at the resulting shapes shown in inset 2 of Fig. 3, we can see how this far-field theory contributes appreciably to our understanding of the resulting optimal geometries. The shapes with the lowest moment and lowest power (with indigo and blue colours) have a concave forward face, to minimise $A^-$. Conversely, the shapes with the highest moment and highest power (with red and orange colours) have a convex or flat forward face, to maximise $A^-$. The unchallenged requirement to minimise $A^+$ results in nearly all shapes having a convex/circular rear face.

Interestingly, the high-moment high-power shapes have similarities to the Edinburgh Duck[5], now commonly referred to as Salter's Duck, which was invented in the 1970s by Stephen Salter. Salter's Duck was intuitively designed to minimise the wave behind the WEC to increase extracted power, and it has been proven to satisfy this goal, but at the cost of high reaction forces[20]. Though not directly comparable because Salter's Duck pitches about a location on the body, unlike our top-hinged WEC, the shape of Salter's Duck appears similar to our high-power high-moment shapes, with curved rearward faces and a longer flat or slightly convex front face. Our optimisation re-discovers the duck-like shape but, importantly, labels it as the optimal shape for maximising power only, and the least optimal shape for minimising PTO moment. An important conclusion from Fig. 3 is that the rate of increase of power along the Pareto Front is half that of the moment. Therefore, by choosing a shape on the Pareto Front that is not the extremely high-power, high-moment shape, we can substantially reduce the reaction moment, without substantially reducing the extractable power. For example, compared to the PTO moment for shape C in Fig. 3, the PTO moment is reduced by 35% for shape B, while still extracting 86% of the power. Furthermore, the PTO moment for shape A is reduced by 48%

compared to shape C, while still extracting 70% of the power. This relation indicates that we can make more reliable (and thus affordable) WECs while still extracting a considerable amount of energy, discovering a new, promising direction of development of WECs.

## Effect of device width: wider devices have smaller motion amplitudes

It is necessary to examine the practicality of each of the optimal solutions resulting from the multi-objective optimisation. Such analyses help to narrow down the set of solutions to an overall optimal solution. Figure 4a shows the pitch response amplitude ($|\xi_5|$, nondimensionalised by $s_1/A$, where $A$ is the amplitude of the incident wave) required to achieve the target power for each shape on the Pareto Front. Larger motions tend to be beneficial for power production, but very large motions can lead to undesirable effects such as over-centring or striking end stops during operation. As shown in Figure 4a, WECs of smaller width ($k_r l = 0.5$) need to move much more than WEC shapes of larger width. In this study, we assume linear potential flow. The inviscid assumption is valid for small (<1) values of the Keulegan–Carpenter (KC) number, the ratio of drag force to inertia force. For waves of relatively small steepness, the KC number for the mid-width WEC ($k_r l = 1.1$) is ~0.4. The KC number decreases further with increasing WEC width and decreasing body motion. Moreover, it has previously been established that nonlinear effects are greater for larger body motion amplitudes[21–23]. As shown in Fig. 4a, the motion amplitudes of the wider ($k_r l = 1.1, 1.5$) WECs are moderate (for relatively low wave amplitudes characteristic of operational seas) and therefore the linear assumption is reasonable. When choosing the width of WEC, there is an opportunity to choose a width large enough to avoid excessive viscous and nonlinear forces. Furthermore, this analysis shows that the potential-flow-based model we use is appropriate for the problem, rather than a complex and computationally expensive high-fidelity model.

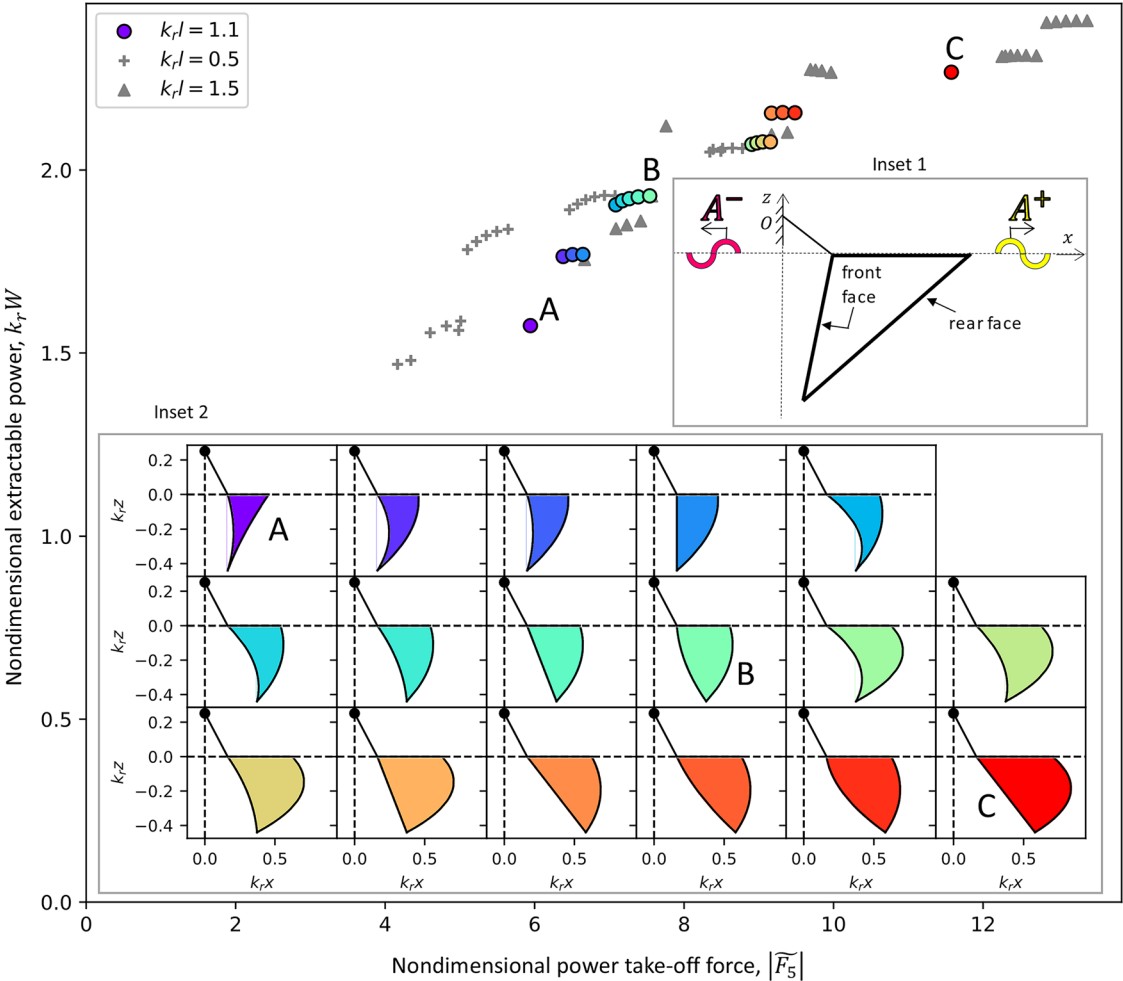

**Fig. 3 | Pareto Fronts of the optimal wave energy converter geometries.** The three widths are represented by different markers: $k_r l = 0.5$ (grey '+' symbols), $k_r l = 1.1$ (coloured circles), and $k_r l = 1.5$ (grey triangles), where $k_r$ is wavenumber and $l$ is the width. The $x$-axis is $|\widetilde{F_5}|$, nondimensionalised PTO reaction moment and $y$-axis is $k_r W$, nondimensionalised extractable power (where $W$ is extractable power over incident power per unit crest length). The colours are used to map data points to the corresponding shapes. Inset 1: Waves made by the WEC when forced to oscillate in otherwise calm water: in the direction opposite to the incident wave ($A^-$, pink) and in the direction of the incident wave ($A^+$, yellow); Inset 2: The differently coloured plots show the 2D cross-section shapes corresponding to the $k_r l = 1.1$ optimisation (outlined circles).

## Applicability of shapes in a wide range of sea states: frequency and direction bandwidths

The optimisations consider idealised monochromatic unidirectional waves of prescribed frequency and direction. In practice, it is necessary to consider how the WEC responds to a range of wave frequencies and directions to ensure the WEC operates properly in typical sea states. For each shape on the Pareto Fronts, we fix the geometry and PTO damping and examine the power, $k_r W$, for prescribed ranges of incident wave (i) frequencies $\omega/\omega_r$ and (ii) directions $\theta$. To characterise the width of the relationship between $k_r W$ and (i) $\omega/\omega_r$ and (ii) $\theta$, we calculate the half-width at half-height, $\Delta_\omega$ and $\Delta_\theta$, for each shape. Larger values of $\Delta_\omega$ and $\Delta_\theta$ correspond to a WEC that works well over a wider range of incident frequencies and directions. The method quantifies the general applicability to a wide range of sea states. Figure 4b shows that the frequency bandwidth is narrower for the smaller widths, suggesting that the larger width shapes perform better over a wider range of incident sea states. Note that we only consider the three widths, and so there may be a particular $k_r l$ value above which the bandwidth no longer increases. Furthermore, low-power low-moment shapes have a slightly wider bandwidth than high-power high-moment shapes. Figure 4c shows that none of the shapes are very sensitive to incident direction (i.e. the bandwidths are all wide), and that device width does not affect this sensitivity.

## Suitability of optimal shapes in a real sea-state

To further examine the suitability of the newly-discovered WEC designs, we analyse power and moment at a real ocean site: the European Marine Energy Centre, a marine energy test site off the coast of the Orkney Islands in Scotland, United Kingdom. We calculate (see 'Methods') the mean annual power and mean annual moment, in addition to the maximum power and moment for a particular occurrence sea-state, for each shape of the mid-width Pareto Front. Shown in Fig. 5, we compare the results for our four metrics for the shapes on the Pareto Front compared to the highest-moment, highest-power shape (shape C in Fig. 3) by showing a percentage difference. From this figure, we see that the trend of moment dropping at least twice as much as power for the other shapes on the Pareto Front is still true for the real sea-state. For example, comparing shape B with shape C from Fig. 3, we calculate that the mean annual power for shape B is 12% smaller than for shape C, but the mean moment is 34% smaller than for shape C. This is further evidence that these shapes could produce high amounts of power for lower moments. We also see that the low-moment, low-power shapes (i.e. shape A) only have slightly (10%) less mean annual power than shape C with substantially (40%) less moment, which was not true for the results on the original Pareto Front. Our hypothesis to explain this result

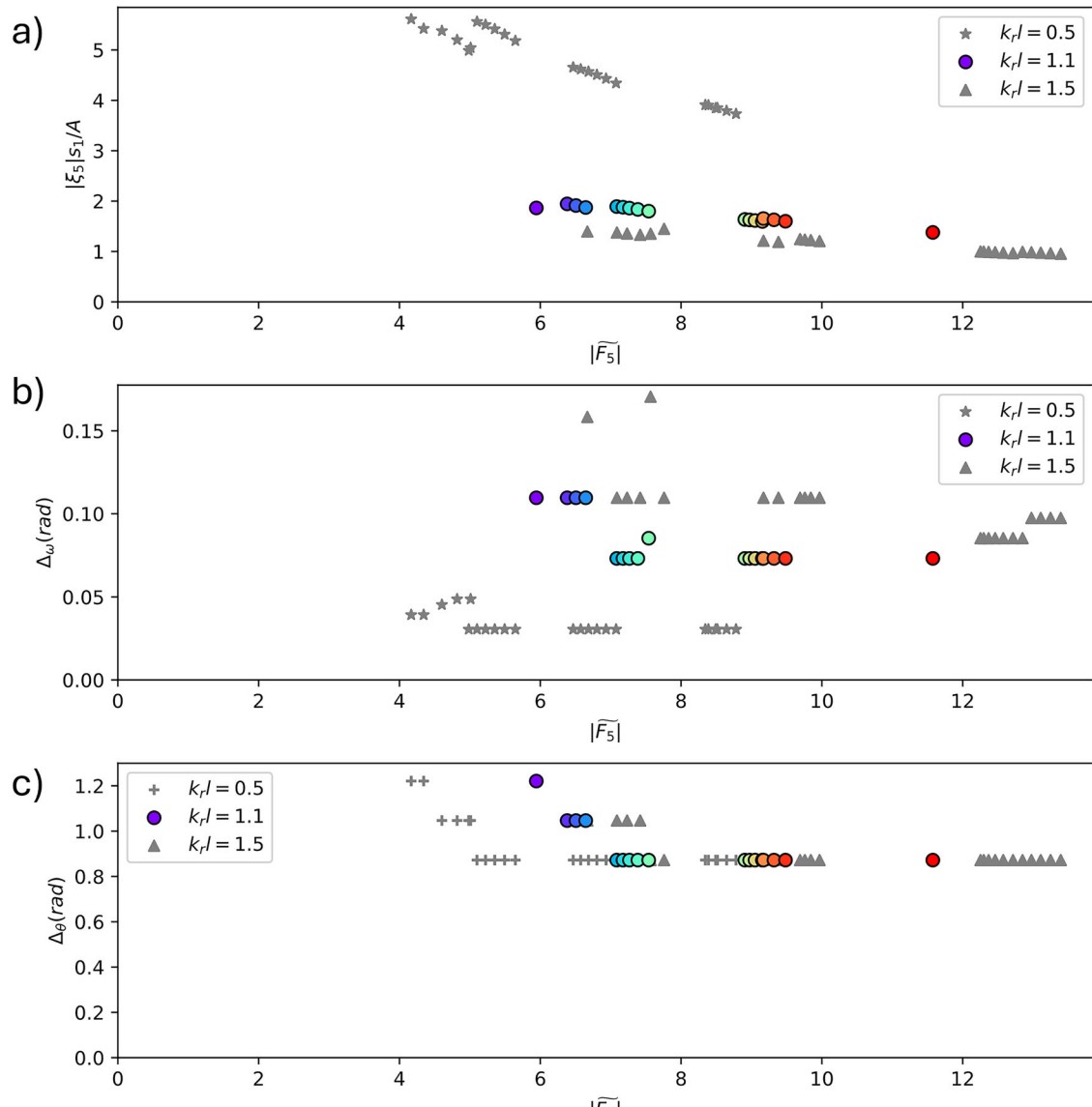

**Fig. 4 | Practicality of optimal shapes on the Pareto Fronts. a** Nondimensionalised pitch body motion response, $|\xi_5|s_1/A$, where $|\xi_5|$ is the amplitude of pitch motion, $s_1$ is the horizontal length of the rigid arm, and $A$ is the incident wave amplitude; **b** Half-width at half-height, $\Delta_\omega$, of the frequency bandwidth for nondimensionalised power, $k_rW$, where $k_r$ is wavenumber and $W$ is the extractable power over incident power per unit crest length; and **c** Half-width at half-height, $\Delta_\theta$, of the direction bandwidth for $k_rW$. All points are plotted against $|\widetilde{F_5}|$, nondimensionalised power take-off force, to map the results to the other figures in this paper. The discrete colours for $k_rl = 1.1$ (where $l$ is device width) are used to map the data to corresponding shapes in Inset 2 of Fig. 3.

is that, as shown in Fig. 4b, the low-moment, low-power shapes have wider frequency bandwidths than the high-power, high-moment shapes. In a real sea-state the bandwidth is an important parameter in determining the power and moment.

## Discussion

We have determined a robust framework to optimise the geometry of a WEC. We present a multi-objective optimisation of the geometry of a pitching top-hinged WEC, which maximises extractable power while minimising the required PTO moment. This has resulted in the discovery of a number of optimal geometries for wave energy extraction. The characteristics of the resulting shapes are consistent with theory pertaining to far-field behaviour of waves that radiate from the body when it is forced to oscillate. Consequently, almost all of the shapes have a convex/circular rear face. The lowest-moment and lowest-power shapes have a concave forward face, whereas the highest-moment and highest-power shapes have a flat forward face.

One of the biggest challenges to wave energy technology is lowering design loads on the structure, without compromising extractable power. Therefore, by considering minimisation of PTO moment as an optimisation function, we enabled the discovery of WEC shapes, which, when compared to the idealised highest-power shape, experience substantially less load (~35%) for only slightly less power (~12%). Furthermore, the lower-load, lower-power shapes have a wider bandwidth of response for different frequencies and directions than the higher-load, higher-power shapes, suggesting that they will achieve high efficiencies over a wider range of sea states. We have shown that the width of the WEC does not affect the geometric characteristics of the optimal shapes, but larger-width WECs have smaller motion amplitudes and wider bandwidths of responses than smaller-width WECs. Finally, we have shown that our conclusions are consistent when considering an example real sea-state off the coast of Scotland, United Kingdom.

We focus on one of the most promising types of WEC: the top-hinged WEC, a class of WECs well-suited for deployment alongside other marine

infrastructure. Hence, the developed shapes and the associated improvements in performance obtained herein are specific to top-hinged WECs. While it is beyond the scope of this paper to examine all potential types of WEC, we note that the methodology could be adapted to other WEC categories (adjusting the framework to suit the different working principles of the different types of device), and this may yield similar improvements in device performance. In general, our results suggest that geometry optimisation is of major importance in the design of a WEC, due to the complex but critical dependence of wave-structure interaction on body geometry, and specifically when considering competing objective functions.

It should be noted that we ignore viscous effects, and so the shapes should ideally not have sharp corners. However, the main characteristics of the geometries and how the geometry affects performance should be consistent when including viscous effects. Further work will include higher-fidelity numerical modelling and physical modelling at laboratory scale, to verify the performance gains predicted using the optimisation approach herein. In particular, the nonlinear hydrostatic restoring force and other nonlinear forces should be investigated. For geometries with rapid change of the waterplane near the waterline, such as shape C, the nonlinear hydrostatic restoring force will become important for moderately large motion response.

Due to the present nascent stage of wave energy technology, and given that the focus of our study is on fundamental hydrodynamics and dependency of performance on body geometry, actual LCOE is not calculated here. Instead, minimisation of PTO moment is used as a proxy for cost reduction that provides a link between LCOE and geometry that is common across all materials. At a later stage, other LCOE factors such as material

weight and maintenance strategy should be investigated to determine the optimal shape from the resulting Pareto Front set.

A top-hinged WEC can be used in isolation or attached to a floating offshore wind turbine to increase overall energy yield while sharing infrastructure and deployment/ maintenance schedules and equipment. Alternatively, these devices could be used in an array of WECs in a future extension to this study. Wave energy is non-concurrent with wind energy, so harnessing wave energy would be an attractive way to diversify the renewable energy resources needed to meet Net-Zero goals. The results from this study could move wave energy substantially closer to becoming economically and practically viable.

## Methods
### Hydrodynamic theory
**Far-field expressions for radiation damping and wave excitation forces.** Figure 2a shows the problem setup, and Table 1 lists the assumptions of the hydrodynamic theory. The WEC consists of a floating body on the water surface and a rigid arm, hinged at a fixed point $O$, restricting its motion to pitch (rotation about point $O$) only. We assume the wave amplitude to be small and the fluid to be ideal, allowing linear potential flow theory to be used, which is a good approximation for most operating conditions. Initially, we assume that the incident wave has given frequency $\omega = \omega_r$, and wavenumber $k = k_r = 2\pi/\lambda_r$, where $\lambda_r$ is the wavelength, and the wave is incident at angle $\theta = 0$ (perpendicular to the device). Once the optimisation is completed and the set of optimal solutions found, we examine sensitivity of the performance of these optimal shapes to wave frequency and incident wave angle. We assume a constant water depth, $k_r H = 5.34$.

Although far-field expressions for radiation damping and wave excitation forces are well-known[24–26], we provide a brief overview of the derivation of the expressions used herein to aid the discussion as to how far-field theory gives physical insight into the optimal geometries. We describe the two-dimensional and three-dimensional problems to compare the idealised case of a 2D WEC of infinite width to the realistic problem of a WEC that is uniform along a given width. In these derivations, we assume the body is freely floating with six degrees of freedom, and a single wave of frequency $\omega$ and angle $\theta$ is incident to the body. The depth is constant at $-H$. The overall velocity potential can be expressed as $\Phi(x, y, z, t) = Re\{\phi(x, y, z)e^{-i\omega t}\}$, where $\nabla^2 \Phi = \nabla^2 \phi = 0$ throughout the fluid. The linearised boundary condition at the free surface is $\omega^2\phi - g\partial\phi/\partial z = 0$ (at $z = 0$), where $g$ is acceleration due to gravity, and at the fluid bottom it is $\partial\phi/\partial z = 0$ (at $z = -H$). On the body surface $S_B$, $\partial\phi/\partial n = v_n$, where $v_n$ is the complex amplitude of the normal velocity of $S_B$ and $\mathbf{n}$ is pointing into the body. The general wave potential $\phi$ can be expressed as $\phi = \phi_I + \phi_d + \phi_r$ where $\phi_I$ is the incident wave potential, $\phi_d$ is the diffraction potential (the disturbance of the incident wave due to the fixed body), and $\phi_r$ is the radiation potential (the wave potential due to the body being forced to oscillate with normal velocity in otherwise calm water). The incident wave potential is given by

$$\phi_I = \frac{gA}{\omega}\frac{\cosh k(z + H)}{\cosh kH}e^{ik(x\cos\theta + y\sin\theta)}, \quad (1)$$

where $A$ is the complex wave amplitude, and $k$ is the wavenumber, defined by the dispersion relation $\omega^2 = gk \tanh kH$. On the body surface,

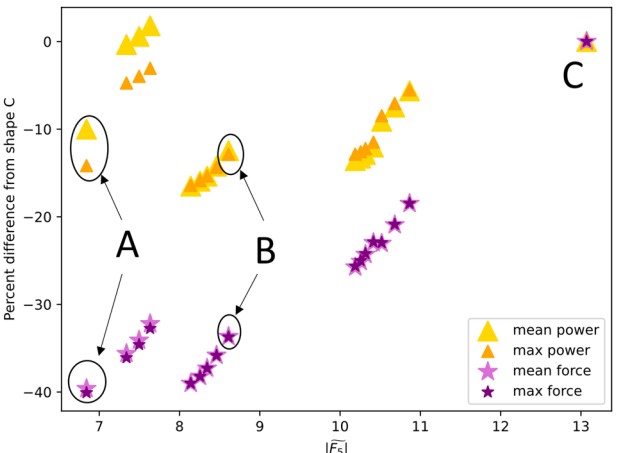

**Fig. 5 | Performance of wave energy converters for a representative location off the coast of Scotland, United Kingdom.** Percent difference of mean annual power (yellow triangles), maximum power from the occurrence matrix (orange triangles), mean annual moment (light purple stars) and maximum moment from the occurrence matrix (dark purple stars) for each shape on the mid-width ($k_r l = 1.1$, where $k_r$ is wavelength and $l$ is device width) Pareto Front, compared to the highest-moment, highest-power shape (shape C in Fig. 3) vs. $|\tilde{F}_5|$, the nondimensionalised power take-off moment at resonance (matching the $x$-axis of the other figures in this paper).

## Table 1 | Assumptions for initial optimisation

| Assumptions |
| --- |
| Linear potential flow theory (wave amplitude is small compared to wavelength; fluid is irrotational and inviscid) |
| Monochromatic unidirectional wave incident from $x = -\infty$ with given frequency $\omega = \omega_r$ |
| Point $O$ is fixed (resulting in pitch motion only about point $O$) |
| Power take-off (PTO) is assumed to be a simple linear damper with a damping coefficient $\beta_{55}$ (which is optimised) |
| Constant water depth with bottom at $z = -H$ |

$\partial\phi_d/\partial n = \partial\phi_I/\partial n = 0$ and $\partial\phi_r/\partial n = \nu_n$. Additionally, the boundary value problems for $\phi_d$ and $\phi_r$ must include a radiation condition. In other words, waves generated by disturbance of the body must be outgoing at infinity. It can be shown that these radiation conditions can be expressed in terms of the Kochin functions as

$$\phi_j = \begin{cases} \frac{-i}{k} H_j \binom{0}{\pi} \frac{\cosh k(z+H)}{\cosh kH} e^{\pm ikx} \text{ as } x \to \pm\infty & \text{2D} \\ -\frac{H_j(\vartheta)}{\sqrt{2\pi kR}} \frac{\cosh k(z+H)}{\cosh kH} e^{\pm ikR + i\pi/4} \text{ as } R \to \infty & \text{3D}, \end{cases} \quad (2)$$

where $j = d, r$, and $(R, \vartheta)$ are polar coordinates about the $z$-axis. $H_j(\vartheta)$ is the Kochin function:

$$H_j(\vartheta) = -\frac{k}{D} \iint_{S_B} \left( \frac{\partial\phi_j}{\partial n} - \phi_j \frac{\partial}{\partial n} \right) \left( \frac{\cosh k(z+H)}{\cosh kH} e^{-ik(x\cos\vartheta + y\sin\vartheta)} \right) dS, \quad (3)$$

where $D = \tanh kH + kH\operatorname{sech}^2 kH = 2\omega/gV_g$, in which $V_g = \partial\omega/\partial k$ is the group velocity, and $j = d, r$. We can see that the Kochin functions describe far-field behaviour of waves due to the motion of the body. These radiation conditions, along with Green's theorem (for full derivation, see ref. [24]), allow the radiation damping coefficient $B_{ij}$ to be expressed in terms of the Kochin functions:

$$B_{ij} = \begin{cases} \frac{\omega\rho D}{2k^2} \left[ H_i(0)H_j^*(0) + H_i(\pi)H_j^*(\pi) \right] & \text{2D} \\ \frac{\omega\rho D}{2\pi k} \int_0^{2\pi} H_i(\vartheta)H_j^*(\vartheta)d\vartheta & \text{3D}. \end{cases} \quad (4)$$

where $\rho$ is the water density. Additionally, the $j$th component of the wave excitation force $X_j$ can also be expressed in terms of the Kochin functions:

$$X_j = \frac{-i\rho gDA}{k} H_j(\pi + \theta). \quad (5)$$

With these expressions for $B_{ij}$ and $X_j$ in terms of the far-field amplitude of the wave produced by oscillating the body in otherwise still water, we now turn to the equation of motion for our defined problem, in which these far-field expressions will be used.

**Extractable power and power take-off moment.** We now overview the equations for the two objective functions in our optimisation: extractable power and PTO moment. We assume that the PTO moment can be modelled as a linear damper:

$$F_5 = \beta_{55}\dot{\xi}_5, \quad (6)$$

where $\beta_{55}$ is the PTO damping coefficient and $\xi_5$ is the body motion in pitch defined about the fixed point $O$. Recalling that the WEC is restricted to motion in pitch only, it can be shown[27] that the equation of motion for the WEC is

$$(I_{55} + A_{55})\ddot{\xi}_5 + (\beta_{55} + B_{55})\dot{\xi}_5 + C_{55}\xi_5 = X_5, \quad (7)$$

where $I_{55}$ is the pitch moment of inertia defined about the fixed point $O$, $A_{55}$ is the pitch added mass defined about the fixed point $O$, $B_{55}$ is the pitch radiation damping defined about the fixed point $O$ and $C_{55}$ is the pitch hydrostatic coefficient defined about the fixed point $O$:

$$C_{55} = \rho g\{S_{11} + V[(z_{CB} - s_2) - (z_{CG} - s_2)]\}, \quad (8)$$

where $S_{11}$ is the waterplane moment, $S_{11} = ((s_1 + r_2)^2 - s_1^3)/3$, $V$ is volume, $z_{CB}$ is the vertical centre of buoyancy, $z_{CG}$ is the vertical centre of gravity, which is set to be $z_{CG} = -0.71H$, to ensure it is sufficiently deep for stability, but within realistic design constraints. In (7), $X_5$ is the pitch excitation moment defined about the fixed point $O$. We can express $\xi_5 = |\xi_5|e^{i\omega t}$

and $X_5 = |X_5|e^{i\omega t}$. Extractable power, averaged over one period, for this WEC is thus

$$P = \frac{1}{2}\beta_{55}\omega^2|\xi_5|^2. \quad (9)$$

Solving (7) for $|\xi_5|$ and substituting it in (9), we get an expanded equation for extractable power:

$$P = \frac{1}{2} \frac{\beta_{55}\omega^2|X_5|^2}{\left[C_{55} - \omega^2(I_{55} + A_{55})\right]^2 + \omega^2(\beta_{55} + B_{55})^2}. \quad (10)$$

To maximise P given a particular shape and frequency, we can control the PTO damping, $\beta_{55}$, and the expression in the first square brackets in the denominator of (10), $[C_{55} - \omega^2(I_{55} + A_{55})]$. Therefore, we take $\partial P/\partial\beta_{55} = 0$ and $\partial P/\partial[C_{55} - \omega^2(I_{55} + A_{55})] = 0$ giving the conditions

$$\beta_{55} = B_{55}, \quad (11)$$

and

$$C_{55} - \omega^2(I_{55} + A_{55}) = 0, \quad (12)$$

which result in optimal power

$$P^{\text{opt}} = \frac{|X_5|^2}{8B_{55}}, \quad (13)$$

which occurs when $|\xi_5^{\text{opt}}|/A = |X_5|/(4\omega B_{55})$ and thus

$$F_5^{\text{opt}} = \frac{|X_5|}{2}. \quad (14)$$

**Far-field expressions for optimal extractable power and PTO moment.** We now substitute the far-field derivations into our equations for extractable power and PTO moment. Putting (4) and (5) (where $i, j = 5$) into (13), we get optimal power in terms of the far-field behaviour of the waves (represented by the Kochin functions):

$$P^{\text{opt}} = \begin{cases} \frac{1}{2}\rho g V_g A^2 \frac{|H_5(\pi)|^2}{|H_5(0)|^2 + |H_5(\pi)|^2} & \text{2D} \\ \frac{\pi}{2} \frac{\rho g V_g A^2}{k} \frac{|H_5(\pi)|^2}{\int_0^{2\pi} |H_5(\vartheta)|^2 d\vartheta} & \text{3D}. \end{cases} \quad (15)$$

It is convenient to define capture width $W$ to be extractable power over incident wave power per unit crest length, $P_I = \rho g A^2 V_g/2$. We non-dimensionalise $W$ with wavenumber $k$. In terms of far-field behaviour, we get

$$kW^{\text{opt}} = \begin{cases} k\left(1 + \frac{|H_5(0)|^2}{|H_5(\pi)|^2}\right)^{-1} & \text{2D} \\ \pi \frac{|H_5(\pi)|^2}{\int_0^{2\pi} |H_5(\vartheta)|^2 d\vartheta} & \text{3D}. \end{cases} \quad (16)$$

This equation is key in using far-field theory to aid in our understanding of how WEC geometry affects performance in terms of maximising extractable power. Equation (16) shows that for the idealised 2D problem, $kW$ is maximised when $A^+ \equiv |H_5(0)|$ is minimised and when $A^- \equiv |H_5(\pi)|$ is maximised. $A^+$ and $A^-$ are represented visually in Fig. 3 in Inset 1. For the 3D problem described in the present study involving a top-hinged WEC, which is uniform in $y$, the conclusions are the same: $kW$ is maximised when $A^+$ is minimised and when $A^-$ is maximised.

To determine PTO moment in terms of far-field behaviour, we can put Eq. (5) into (14):

$$|F_5|^{\text{opt}} = \frac{\rho\omega V_g}{k}|H_5(\pi)|. \quad (17)$$

This equation is the other key part in using far-field theory to determine how WEC geometry affects performance, in terms of minimising PTO moment. Equation (17) shows that the PTO moment is minimised when $A^- \equiv |H_5(\pi)|$ is minimised. Therefore, from Eqs. (16) and (17), we see that our multi-objective optimisation with objective functions that (i) maximise $kW$ and (ii) minimise $F_5$, corresponds to an unchallenged goal to minimise $A^+$. However, there are competing goals between the two objective functions to maximise $A^-$ to achieve goal (i) and minimise $A^-$ to achieve goal (ii). Implications for WEC geometry are discussed in the main text.

### Geometry definition

Figure 2a shows the geometric parameters of the WEC. Here, $s_1$ and $s_2$ define the horizontal and vertical components of the length of the hinge arm, above the water surface. $r_2$ is the length at the waterline between the front and rear faces, and $r_1$ is the horizontal distance from the hinge to the bottom edge. The WEC is uniform in $y$, with width $l$, centred at $y = 0$. Front and rear faces are defined by curves $c_1$ and $c_2$, respectively. These curves are described by basis functions, whereby the coefficients $\overline{a_n}$ and $\overline{b_n}$ of the functions are parameters which determine the geometry, and are thus to be optimised. This method to define geometry enables many, very general geometries to be described by relatively few parameters. In this study, we assume both curves are described by second-order functions.

We use Chebyshev polynomials of the first kind, $T_n$, to define curves $c_1$ and $c_2$. $T_n$ is defined by the recurrence relation: $T_0(x) = 1$; $T_1(x) = x$; $T_{n+1}(x) = 2xT_n(x) - T_{n-1}(x)$, where $x \in (-1, 1)$. The coefficients $a_0$, $a_1$, $b_0$ and $b_1$ are calculated to enable the shape to be represented by $r_1$, $r_2$ and $h$ (shown in Fig. 2a). Coefficients $a_2$ and $b_2$ are the second-order coefficients of $c_1$ (front face) and $c_2$ (rear face), respectively. In this study, we only consider up to second-order terms, so the geometry is completely defined by $[r_1, r_2, h, a_2, b_2]$.

Sensitivity studies were performed to ascertain how different geometric parameters affect power and moment, from which it was determined to set $k_r s_1 = 0.16$, $k_r s_2 = 0.25$, and $k_r h = 0.44$. The $k_r s_1$ and $k_r s_2$ parameters were set constant for practical constraints, usually determined by the support structure. $k_r h$ was set to a constant value because both the power and surface area were found to increase monotonically as depth increased. Having established this relation, the geometric parameters considered in the optimisation are $r_1$, $r_2$ and curves $c_1$ and $c_2$. These parameters were found to influence power and moment in a meaningful (non-monotonic) way, and most clearly demonstrated the effect of body geometry on performance. The PTO mechanism is assumed to be a simple linear damper in pitch located at point $O$.

For any shape considered within the optimisation, it is necessary to determine the hydrodynamic coefficients (i.e. added mass, radiation damping, and excitation force) in order to compute the power and moment. The frequency-domain panel method WAMIT[28] is used to obtain values for the hydrodynamic coefficients about fixed point $O$. We manually define the mesh for the shape, using Python, to input into WAMIT. We identify the smallest arclength of the shape and specify that there are $N_l$ panels along this arclength. Then, the number of panels along the other sides of the shape are determined such that the panels are as close to square as possible. We run WAMIT using this mesh, and then repeat this run, increasing $N_l$ by one. If the hydrodynamic coefficients are not converged to within 3%, we repeat the process of increasing $N_l$ by one and running WAMIT again, until convergence is achieved. This procedure is implemented for each point in the population as well as for the results from Fig. 4.

### Optimisation procedure

We have shown that the device should be in resonance and the PTO damping coefficient should be equal to the radiation damping coefficient at resonance to maximise the extractable power for a WEC moving in a single degree of freedom. These two criteria are enforced herein. To achieve resonance for a given frequency, the pitch moment of inertia is adjusted. If resonance is not possible (i.e. if the pitch moment of inertia

would be required to be negative), the individual is removed from the population.

Figure 2b, c summarises the optimisation procedure. The optimisation procedure is based on a classic Multi-Objective Evolutionary Algorithm approach, as described in ref. 19. Firstly, an initial population is defined, whereby for each individual, the shape is determined by the length parameters and Chebyshev polynomials. For each shape, the hydrodynamic coefficients are found using WAMIT to find the extractable power and PTO moment from (13) and (14), respectively. To achieve maximum power, Eq. (12) is used to find $I_{55}$:

$$I_{55} = \frac{C_{55}}{\omega^2} - A_{55} \tag{18}$$

If the shape is unable to achieve resonance (due to the radiation of gyration being negative), the individual is discarded. The initial Pareto Front is found from the initial population. In a multi-objective optimisation, an individual is said to dominate another individual if it is strictly better in one objective function and no worse in another objective function. The Pareto Front is defined to be the set of all non-dominated solutions in a population. Then, a standard multi-objective evolutionary optimisation algorithm, programmed in Python, based on the theoretical procedure from ref. 19, is run, whereby at each time step, any individual unable to achieve resonance is removed from contention. We performed a sensitivity study for the optimisation procedure to determine the best initial population size, number of generations needed, and mutation probability.

In the first part of this study, a single, monochromatic, unidirectional, unit-amplitude wave is assumed incident upon the body. All lengths are nondimensionalised by the wavenumber $k_r$ of this incident wave; this means that the device can simply be scaled to perform similarly for a different wave frequency. The PTO damping for each shape is different because the optimal PTO damping is equal to the radiation damping at resonance (as shown in Eqs. (11) and (12)).

To determine the frequency and direction bandwidths, the shape is fixed and the PTO coefficient is fixed at the value calculated to maximise power at $\omega_r$, $B_{55}(\omega_r)$. WAMIT is run to find the hydrodynamic coefficients for a range of frequency $\omega$ values about $\omega_r$ and direction $\theta$ values about 0, and power and moment are calculated using Eqs. (13) and (14) for each $\omega$ and $\theta$.

### Keulegan–Carpenter number

The KC number is the ratio of the drag force to the inertia force. In this problem,

$$KC = \frac{VT}{L} = \frac{2\pi \left( \frac{|\xi_5| s_1}{A} \right) (k_r A)}{k_r l}, \tag{19}$$

where $\frac{|\xi_5| s_1}{A}$ is nondimensional pitch response amplitude (as shown in Fig. 4a), $k_r A$ is wave steepness, and $k_r l$ is nondimensional WEC width. For a small wave steepness ($k_r A \sim 0.05$), which is a good assumption for most operational waves, for the mid-width WEC ($k_r l = 1.1$), the pitch response amplitudes are ~1.5 (Fig. 4a), and so the KC number is ~0.4. Larger-width WECs have smaller pitch response amplitudes, and Eq. (19) indicates that the KC number will decrease.

### Determination of power for the example real sea-state

We examine and discuss the suitability of the newly-discovered WEC designs for an actual sea-state observed at the European Marine Energy Centre[29]. To calculate the mean power, we use an occurrence matrix, which shows the percentage of time a sea-state has a particular significant wave height and peak period. The ERA-5 dataset is used for a location with coordinates (59° N, 2.5° W), which is the closest ERA-5 model analysis point, for years 1979–2020 inclusive[30].

$f_{ij}$ is the percent occurrence for the $i$th wave height class and $j$th period class. To calculate the mean power, we take

$$\overline{P} = \sum_{j=1}^{M} \sum_{i=1}^{N} f_{ij} P_{ij} \Delta T, \qquad (20)$$

where $M$ is the number of wave periods in the occurrence matrix, $N$ is the number of wave heights, $P_{ij}$ is the power due to the $i$th wave height class and $j$th period class, and $\Delta T$ is the wave period step. To find $P_{ij}$, we take

$$P_{ij} = 2 \int_0^{\infty} S_{ij} P_5 \, df, \qquad (21)$$

where $P_5$ is power per unit amplitude, and $S_{ij}$ is the spectrum for the $i$th wave height class and $j$th period class. In the present study, we use a Pierson–Moskowitz spectrum:

$$S_{ij}(f) = \frac{H_i^2}{4} (1.057 f_j)^4 f^{-5} e^{\frac{-5}{4}\left(\frac{f_j}{f}\right)^4}, \qquad (22)$$

where $f_j = 1/T_j$ is the peak frequency in bin $j$ and $H_i$ is the significant wave height in bin $i$. To find the mean moment, a similar procedure is followed except that the formula for $F_{ij}$, the moment due to the $i$th wave height class and $j$th period class, is

$$F_{ij} = \int_0^{\infty} \sqrt{2 S_{ij}} F_5 \, df. \qquad (23)$$

For these calculations, for each shape on the Pareto Front, we fix the geometry and PTO damping coefficient and calculate $P_5$ and $F_5$ for each frequency in the spectrum. The PTO is determined to be the radiation damping for the peak frequency, and the shape dimensions are determined from the peak wavenumber. For the maximum power and moment, shown in Fig. 5, we show the maximum $P_{ij}$ and $F_{ij}$, respectively.

## Data availability

The data used for findings summarised in 'Suitability of optimal shapes in a real sea-state' in the main text, and explained in 'Determination of power for the example real sea-state' in the 'Methods' section, are from ERA-5 (an open-access dataset).

## Code availability

The code for this research can be found on GitHub at https://github.com/emmae0/WEC_MOEA.

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

## Acknowledgements

This work was funded by the Engineering and Physical Sciences Research Council, UK, through the 'Supergen ORE Hub' grant (EP/S000747/1). Dr E Barbour is gratefully acknowledged for his valuable advice on the manuscript.

## Author contributions

E.C. Edwards: conceptualisation, methodology, validation, formal analysis, writing—original draft, visualisation, project administration, funding acquisition; C. Whitlam: conceptualisation, methodology, validation, writing—review and editing, supervision; J. Chapman: conceptualisation, methodology, writing—review and editing, supervision, funding acquisition; J. Hughes: conceptualisation, methodology, validation, writing—review and editing; B. Redfearn: conceptualisation, methodology; S. Brown: conceptualisation, writing—review and editing, funding acquisition; S. Draper: writing—review and editing, methodology; A. Borthwick: writing—review and editing; G. Foster: conceptualisation, supervision, funding acquisition; D.K.P. Yue: conceptualisation, writing—review and editing, supervision; M. Hann: conceptualisation, writing—review and editing, supervision, funding acquisition; D. Greaves: conceptualisation, writing—review and editing; supervision, funding acquisition

## Competing interests

C.W., J.C., J.H., B.R. and G.F. are/ have been employed by Marine Power Systems. All other authors declare no competing interests.
