## [Transparent Peer Review file · Communications Engineering]

The effect of device geometry on the performance of a wave energy converter

Corresponding Author: Dr Emma Edwards

Version 0:

Reviewer comments:

Reviewer #1

(Remarks to the Author)

The authors investigated the effect of the geometry of the WEC on power absorption and PTO torque. While numerous studies have explored the effect of WEC geometry on power absorption, making it a well-trodden research topic, this paper offers a degree of novelty. This paper can be said to summarize many studies on the shape of WEC, and investigates the balance between power absorption and PTO torque, contributing valuable insights to the field.

Meanwhile, authors need to answer the following questions:

- 1) The colors of A+ and A- in inset 1 of Figure 3 are not related to the colors in inset 2. Different colors should be set to distinguish them and avoid increasing the difficulty of reading the figure.
- 2) The vertical axis of Figure 4 b) should have a unit.
- 3) The content in the appendix applies to WECs without hinge points, only pitching around their own axis, as in equations (A7) to (A14). When the WEC has hinge points, some equations cannot be directly used. The author seems to have only performed frequency domain calculations and not time domain calculations, so the hydrodynamic calculations in this paper do not actually consider the impact of hinges on the WEC. That is, the research subject of this paper is actually a WEC pitching around its own axis without a hinge point, rather than a hinged WEC.

Reviewer #2

(Remarks to the Author)

Geometry is one of the most important factors of wave energy converter's performance. However, the discussion in this manuscript does not provide an in-depth and broad analysis. Therefore, I cannot recommend it for publication. Some suggestions are given below

- The design of WEC covers a wide range such as point absorbers, hinged system, oscillating water column, and over topping devices. However, this manuscript only covers a specific top hinged device. Particularly, quite a few authors have studied different designs.
- While the authors have stated about LCOE, factors related to cost shall be included such as material weight and maintenance strategies related to geometry.
- As sea state is mentioned, optimal scale shall be studied.
- Furthermore, array is the final setup for all offshore renewable system. With that, the parameters of arrays such as relative distance or direction might be discussed together with geometry.

Overall, this is a good study but it seems to be more suitable for a specialized journal such as renewable energy.

Reviewer #3

(Remarks to the Author)

In this work, the authors presented an optimisation of the geometry of the device to maximise and minimize power for different conditions. But it is not clear how to find the maximum and minimum power. Calculation implementation is not clear.. In the appendix, the method and power extracted are described very well. and Eq A11, why radiation damping and PTO damping should be equal? Generally, the paper is well written and can be accepted if the results of power against different

conditions (wave periods, wave heights, different PTO damping) and its calculation details should be explained.

Version 1:

Reviewer comments:

Reviewer #1

(Remarks to the Author)

The reviewer's suggestion for this manuscript is reject. The comments are as follows,

- 1) The authors investigated the effect of the geometry of the WEC on power absorption and PTO torque. In the past, many researchers have studied the effect of the geometry of WEC, the authors merely used a Multi-Objective Evolutionary Algorithm (MOEA) approach to study more shapes. The research in this paper is not innovative enough and lacks breakthrough.
- 2) The research content of this paper is not complete and rich enough, and there is a lack of physical model experiments in a wave tank and sea trials to verify the accuracy of the research results.
- 3) The literature review section of this paper reviewed and cited too few relevant papers, and the authors did not comprehensively and accurately review existing research. The author's review of optimization studies on the geometry of WEC is also not thorough enough.

Reviewer #3

(Remarks to the Author)

Revised version can be accepted.

Reviewer #4

(Remarks to the Author)

In this paper, a parametric study of a Salter duck-type wave energy converter was performed with focus on geometrical shape optimization with respect maximum power and minimum PTO force. The paper is in general well written. I was asked to review the revision. I don't have many comments. Some minor comments are provided below. In general, it will be appreciated if the limitations or the assumptions of the hydrodynamic theory and the motion analysis were provided for example in a table. The geometrical shape of a the 2D WEC only uniquely determines the hydrodynamic properties, while the inertia of moment and the stiffness term for the pitch motion also depends on the mass distribution of the floater. Some information about the mass distribution of the floater needs to be provided and predefined. The authors did not mention the viscous damping effect due to the drag force acting on the floater and the justification of such simplification should be made. The obtained geometrical shapes show a rapid change of the water plane areas when there are significant pitch motions, which may first lead to a nonlinear effect of hydrostatic restoring effect. Some general comments about neglecting such effects should be provided.

Manuscript No.: COMMS-24-0277-T

Title: The effect of device geometry on the performance of a wave energy converter

Authors: Emma C Edwards, Craig Whitlam, John Chapman, Jack Huges, Bryony Redfearn, Scott Brown, Scott Draper, Alistair GL Borthwick, Graham Foster, Dick K-P Yue, Martyn Hann, Deborah Greaves

Authors' Response to Reviewer #1:

The authors investigated the effect of the geometry of the WEC on power absorption and PTO torque. While numerous studies have explored the effect of WEC geometry on power absorption, making it a well-trodden research topic, this paper offers a degree of novelty. This paper can be said to summarize many studies on the shape of WEC, and investigates the balance between power absorption and PTO torque, contributing valuable insights to the field.

Response: Thank you for reviewing our manuscript. We very much appreciate your comments. Our response to your comments is given below, with line numbers according to the marked-up PDF. Changes to the manuscript are highlighted in blue font.

Meanwhile, authors need to answer the following questions:

1) The colors of A+ and A- in inset 1 of Figure 3 are not related to the colors in inset 2. Different colors should be set to distinguish them and avoid increasing the difficulty of reading the figure.

Response: Following the suggestion, we have changed the colours of A+ and A- in Inset 1 of Figure 3 to be pink and yellow, which are different to the colours used in the Pareto Front and in Inset 2.

2) The vertical axis of Figure 4 b) should have a unit.

Response: Done.

3) The content in the appendix applies to WECs without hinge points, only pitching around their own axis, as in equations (A7) to (A14). When the WEC has hinge points, some equations cannot be directly used. The author seems to have only performed frequency domain calculations and not time domain calculations, so the hydrodynamic calculations in this paper do not actually consider the impact of hinges on the WEC. That is, the research subject of this paper is actually a WEC pitching around its own axis without a hinge point, rather than a hinged WEC.

Response: The equations used relate to a WEC hinged about a rigid fixed point. In the revised manuscript, we clarify that the hydrodynamic coefficients, hydrostatic stiffness and pitch moment of inertia are defined about the fixed point O throughout section A.1.2. We also state that point O was specified in WAMIT as the reference point about which the hydrodynamic coefficients are defined; see line 402.

Authors' Response to Reviewer #2:

This is an interesting study. Comments are as follows:

Response: Thank you for reviewing our manuscript. We very much appreciate your comments. Our response to your comments is given below, with line numbers according to the marked-up PDF. Changes to the manuscript are highlighted in blue font.

Geometry is one of the most important factors of wave energy converter's performance. However, the discussion in this manuscript does not provide an in-depth and broad analysis. Therefore, I cannot recommend it for publication. Some suggestions are given below.

- The design of WEC covers a wide range such as point absorbers, hinged system, oscillating water column, and over topping devices. However, this manuscript only covers a specific top hinged device. Particularly, quite a few authors have studied different designs.

Response: It would be impossible to do a geometry optimisation of all types of WECs due to the differing working principles for each type of WEC. We also note that while other studies have compared multiple categories of WEC, we are not aware of any that have done so in the context of geometry optimisation. To our knowledge, this study is the first to consider how WEC geometry affects PTO force or how geometry can be exploited to minimise PTO moment. Given that geometry is fundamental to wave-structure interaction, this addresses a significant gap in current knowledge. The results from our work (i.e., novel shapes that significantly lower PTO moment while only slightly lowering extractable power) demonstrate the proficiency of the framework and methodology we developed. The methodology could be adapted to other types of WECs and it is likely that similar improvements could be found, but it is outside the scope of the present study to consider these other types of WECs.

In the revised manuscript, we have added more contextual information about our choice of a particular type of WEC (lines 49-63). We also added an explanation in the introduction about how it is not feasible to perform a simultaneous geometry optimisation for multiple types of WECs (lines 83-87). In the discussion, we added a paragraph on how the methodology could be adapted to other types of WECs and how our study shows the crucial dependence of WEC performance on body geometry (lines 251-259).

- While the authors have stated about LCOE, factors related to cost shall be included such as material weight and maintenance strategies related to geometry.

Response: We have added a discussion (lines 264-270) about LCOE in the revised manuscript. Please note that the focus of our study is on fundamental hydrodynamics and

the dependency of performance on body geometry. The findings are common for all materials, and so to maximise generality actual materials are not specified. We have added a note into the revised manuscript to state that after performing a general optimisation, such as the one included herein, the results would of course have to be adapted to a specific construction before a device could be deployed at sea. However, this is beyond the scope of our manuscript. In short, material specification would limit the generality of the conclusions. This is also true of maintenance.

- As sea state is mentioned, optimal scale shall be studied.

Response: In lines 429-434 of the revised manuscript we state that the results in sections 2-3 are based on a single, monochromatic, unidirectional, unit-amplitude wave incident upon the body (not a sea-state). We have added discussion about how the results are nondimensionalised by the wavenumber so that the body can be scaled to any particular incident wave. These assumptions are also stated in the main text, in lines 127-129 of the revised manuscript. For clarification, we note that for the real sea-state, the dimensional values of the length dimensions are determined by the peak frequency (lines 463-465).

- Furthermore, array is the final setup for all offshore renewable system. With that, the parameters of arrays such as relative distance or direction might be discussed together with geometry.

Response: These types of devices could be used in isolation or attached to a floating wind turbine. They could also form part of an array of WECs, and so we have added a sentence to the discussion (see lines 273) indicating that our study could be extended to examine arrays of such devices. While we agree that the design optimization of arrays of WECs is likely to be a promising area of research in future, our focus is more near-term and hence we leave such work to future studies.

Overall, this is a good study but it seems to be more suitable for a specialized journal such as renewable energy.

Response: *Communications Engineering* has a broad scope, including marine and ocean engineering, which we believe our manuscript fits well. Furthermore, the journal places an emphasis on UN Sustainable Development Goals (SDGs), and our manuscript aims to progress SDG7: Affordable and Clean Energy. It is well established that wave energy technology is not as well developed as other renewable energy technologies, such as wind and solar energy devices. Advances in wave energy converter design are therefore important in achieving SDG7. Moreover, a diverse set of renewable energy sources and technologies is needed to meet SDG and Net-Zero goals globally due to the intermittent

nature of all renewable energy. Importantly, wave energy is not niche in terms of global resource—it has been shown that there is enough energy in ocean waves to meet the entire global energy demand. Almost three-quarters of the world's population live within 50 km of the sea. Wave energy is a vast but as-yet untapped form of renewable energy. Given that wave energy technology is relatively nascent, improvements in knowledge and performance arguably are likely to be more impactful than incremental increases for more established technologies such as wind and solar cell energy devices. The novel framework and results presented in our manuscript, which indicate potential improvements of the order of 40%, could yield significant cost improvements in wave energy technology and bring it closer to being more cost-competitive with more established offshore renewable energy technologies.

Authors' Response to Reviewer #3:

In this work, the authors presented an optimisation of the geometry of the device to maximise and minimize power for different conditions.

Response: Thank you for reviewing our manuscript. We very much appreciate your comments. Our response to your comments is given below, with line numbers according to the marked-up PDF. Changes to the manuscript are highlighted in blue font.

But it is not clear how to find the maximum and minimum power. Calculation implementation is not clear.

Response: We have clarified, in line 419 of the revised manuscript, that equations A13 and A14 are used to calculate power and PTO moment. To maximise power and minimise PTO moment, a multi-objective optimisation is implemented. See lines 415-416, lines 424-428, and figure 2 in the revised manuscript.

In the appendix, the method and power extracted are described very well. and Eq A11, why radiation damping and PTO damping should be equal?

Response: In lines 347-350 of the revised manuscript, we provide clarification that the condition is found by taking the derivative of the power equation with respect to the PTO damping coefficient and the hydrostatic and mass term (resonance condition).

Generally, the paper is well written and can be accepted if the results of power against different conditions (wave periods, wave heights, different PTO damping) and its calculation details should be explained.

Response: We thank the reviewer for their supportive comment. In lines 429-434 of the revised manuscript, we emphasise that the results in sections 2 and 3 are based on a single monochromatic unidirectional unit-amplitude wave incident upon the body (not a sea-state). We have added discussion about how the results are nondimensionalised by the wavenumber so that the body can be scaled to any particular incident wave. The assumptions are stated in lines 127-129.

For clarification purposes, we have added that for the real sea-state, the dimensional values of the length dimensions and the PTO damping are determined by the peak frequency (see lines 463-465 of the revised manuscript). We have also added more clarification in lines 435-438 stating that section 4 considers a range of frequency and direction values (not a particular sea-state) and that the PTO damping is fixed. This is also mentioned in lines 200-202 of the main text of the revised manuscript.

Manuscript No.: COMMSENG-24-0277A

Title: The effect of device geometry on the performance of a wave energy converter

Authors: Emma C Edwards, Craig Whitlam, John Chapman, Jack Huges, Bryony Redfearn, Scott Brown, Scott Draper, Alistair GL Borthwick, Graham Foster, Dick K-P Yue, Martyn Hann, Deborah Greaves

Dear Editor,

Thank you for your email concerning our manuscript titled “The effect of device geometry on the performance of a wave energy converter” (COMMSENG-24-0277A) which was submitted to *Communications Engineering* for possible publication. We have modified the manuscript taking full consideration of all the comments and suggestions from the Reviewer and yourself. Corresponding revisions are indicated by blue font in the new version of our manuscript. We also provide itemized responses to each of the referee’s comments and each of your comments. Two copies of the pdf are submitted: one where changes to the manuscript are indicated in blue, and the other the final version.

We would like to thank you and the anonymous reviewer for the detailed critique of our manuscript.

With best regards.

Sincerely yours,

Dr Emma Edwards

Authors' Response to Reviewer #4:

In this paper, a parametric study of a Salter duck-type wave energy converter was performed with focus on geometrical shape optimization with respect maximum power and minimum PTO force. The paper is in general well written. I was asked to review the revision. I don't have many comments. Some minor comments are provided below.

Response: Thank you for reviewing our manuscript. We very much appreciate your comments. Our response is given below, with line numbers according to the marked-up PDF. Changes to the manuscript are highlighted in blue font.

In general, it will be appreciated if the limitations or the assumptions of the hydrodynamic theory and the motion analysis were provided for example in a table.

Response: We have added a description (lines 104-107) of the main hydrodynamic assumptions in the main text and we have also added a table (page 10) that summarizes the assumptions in the Methods section.

The geometrical shape of a the 2D WEC only uniquely determines the hydrodynamic properties, while the inertia of moment and the stiffness term for the pitch motion also depends on the mass distribution of the floater. Some information about the mass distribution of the floater needs to be provided and predefined.

Response: In our optimisation, the pitch moment of inertia is determined to ensure that the body is in resonance. This is explained in the Methods section (lines 409-414), where we explicitly state the equation for pitch moment of inertia (equation A18, line 412). The pitch hydrostatic stiffness term is given in Equation A8 (line 332). We have clarified that the centre of gravity is fixed (and stated it) and chosen its value to ensure the centre of gravity is sufficient for stability, but within realistic design constraints (lines 334-335).

The authors did not mention the viscous damping effect due to the drag force acting on the floater and the justification of such simplification should be made. The obtained geometrical shapes show a rapid change of the water plane areas when there are significant pitch motions, which may first lead to a nonlinear effect of hydrostatic restoring effect. Some general comments about neglecting such effects should be provided.

Response: We have added a sentence in the second section when defining the assumptions (lines 106-107) to indicate that the linear potential flow assumption is discussed in the 'Effect of device width' section. Here, we comment on both viscous effects and nonlinear effects (lines 167-177). We also discuss the KC number while explaining that the viscous force component is smaller than the inertial force component for our bodies. We note why nonlinear effects are greater for larger motion amplitudes. We have added a further sentence (lines 172-174) to emphasise that the motion amplitudes of the WECs are moderate (for relatively low wave amplitudes characteristic of operational seas) and hence the linear

assumption is reasonable. We also mention (lines 240-242) that we ignore viscous effects, and so the shapes should ideally not have sharp corners but that it does not affect our results. We comment (lines 244-247) on the importance of the nonlinear hydrostatic force for relatively large motion responses of geometries with rapid change of waterplane near the waterline, and mention that a next step in our study would be to investigate such forces using higher-fidelity modelling.

Authors' Response to Editor:

Every figure needs a short title which does not refer to specific panels within the figure itself. Please ensure all non-standard abbreviations used in figures and legends are defined in each figure in which they are used.

Response: We have added an introductory sentence to each figure (Figure 2 on page 4, Figure 3 on page 5, Figure 4 on page 7 and Figure 5 on page 8). We have also inserted definitions into the captions of the abbreviations used in the figures.

You'll need to provide a link to Github for readers to access the code.

Response: We have now added the link to Github (line 278).

We don't publish Appendices, instead create Supplementary Information files. Each SI item should be labelled (e.g. SI note 1, SI figure 2) and please refer to them in the main text.

Response: We confirm we do not have any appendices or supplementary information files.

You'll need to include a competing interests statement

Response: We confirm have added a competing interests statement (lines 279-281)

You'll need to include an author contributions statement

Response: We confirm we have included an author contribution statement (lines 264-273).

We don't use numbering for sections. This will be removed by the Production team. So if you refer to section numbering in the main text, you'll need to remove that.

Response: We confirm that we do not refer to section numbering in the main text.